# Clinical Course from Diagnosis to Death in Patients with Well-Differentiated Thyroid Cancer

**DOI:** 10.3390/cancers12082323

**Published:** 2020-08-18

**Authors:** Hyunju Park, Jun Park, So Young Park, Tae Hyuk Kim, Sun Wook Kim, Jae Hoon Chung

**Affiliations:** 1Division of Endocrinology and Metabolism, Department of Medicine, Samsung Medical Center, Sungkyunkwan University School of Medicine, Seoul 06351, Korea; hj1006.park@samsung.com (H.P.); jun113.park@samsung.com (J.P.); taehyukmd.kim@samsung.com (T.H.K.); swkimmd@skku.edu (S.W.K.); 2Division of Endocrinology, Department of Medicine, Korea University Ansan Hospital, Ansan 15355, Korea; psyou0623@naver.com

**Keywords:** cause of death, well-differentiated thyroid cancer, clinical course, overall survival, distant metastasis

## Abstract

Because of the low mortality rate of well-differentiated thyroid cancer (WDTC), investigation of the clinical course leading to death is limited. We analyzed the cause of death and clinical course from diagnosis to death in patients who died of WDTC. A total of 592 WDTC patients died between 1996 and 2018. After exclusion, 79 patients were enrolled and divided into four groups based on their clinical course; that is, inoperable at the time of diagnosis (inoperable), distant metastasis (DM) detected at the time of diagnosis (initial-DM), DM detected during follow-up (late-DM), and loco-regional disease (L-R). Lung (55.6%) in papillary thyroid carcinoma (PTC) and bone (46.7%) in follicular thyroid carcinoma (FTC) were the most common metastasis locations. The most common causes of death were respiratory failure (32.3%) and airway obstruction (30.6%) in PTC, and complications due to immobilization arising from bone metastasis (35.3%) in FTC. Brain metastasis was found in 13.3% of patients and had the worst prognosis. The overall survival (OS) differed significantly (*p* = 0.001) according to clinical course; the inoperable had the shortest survival, followed by the initial-DM, L-R, and late-DM. However, OS did not differ significantly between PTC and FTC patients with initial-DM (*p* = 0.83). Other causes of death were far more common than death resulting from WDTC. In patients dying of WDTC, the major cause of death varied by metastatic site. OS differed according to clinical course, but not histologic type. Timing and DM sites differed between PTC and FTC.

## 1. Introduction

Papillary thyroid carcinoma (PTC) and follicular thyroid carcinoma (FTC) are referred to as well-differentiated thyroid carcinoma (WDTC), which arises from thyroid follicular cells [1]. PTC and FTC comprise approximately 85% and 11% of all thyroid cancer cases, respectively [2]. Thyroid cancer incidence has been increasing in many countries, mainly owing to early detection of PTC <1 cm in diameter with high-resolution ultrasonography (US). However, the mortality of WDTC has not changed [3,4], and disease-specific mortality at 10 years is less than 5% [5,6].

Most WDTCs are slow growing and have favorable outcomes; regardless, some patients die of thyroid cancer. Among many prognostic factors [7,8,9], distant metastasis is an important factor that increases the mortality rate of WDTC [10,11,12]. Lung is the most common site of distant metastasis, followed by bone [13]. However, it is difficult to investigate the clinical course from diagnosis to death in patients who die of WDTC because of its indolent clinical course and low mortality rate.

While both FTC and PTC comprise WDTC, the epidemiologic and clinical behavior of FTC differs from that of PTC [14]. Loco-regional lymph node metastasis is common in PTC, whereas distant metastasis is more common in FTC.

Previous studies described a limited number of lethal cases of WDTC and did not evaluate PTC and FTC separately. In this study, we investigated the clinical course from diagnosis to death and the cause of death in patients who died of WDTC. We also compared the clinical features of PTC with those of FTC.

## 2. Results

### 2.1. Causes of Death in 483 Patients Who Died with WDTC

Among 483 patients, 296 patients (83.6%) died of various diseases other than WDTC, while 79 patients (16.4%) died of WDTC (Figure 1). Of the 296 patients, 212 died of other advanced primary cancers; WDTC was in complete remission in most of those patients, and even if the WDTC remained, the extent of disease was local, and did not progress further other primary cancer; 20 died of cardiovascular disease, 20 died of brain hemorrhage or neurologic disease, 40 died of other causes (chronic kidney disease, asthma, pneumonia, infection, liver failure, rheumatoid disease, chronic pulmonary disease, and accident), and 4 died of immediate postoperative complications after surgery for WDTC. The remaining 108 patients died of diseases other than WDTC, but the cause of death was unknown (Table 1).

### 2.2. Clinicopathologic Characteristics of 79 Patients Who Died of WDTC

The median patient age at diagnosis was 67.0 years (interquartile range, 57.5–72.5), and 52 patients (65.8%) were female (Table 2). The median follow-up duration was 64 (32–91) months. All patients had distant metastases or loco-regional invasion. Distant metastases were detected in 57 patients (72.2%), including 33 patients at the time of diagnosis (initial-DM group) and 24 patients during follow-up (late-DM group). Sixteen patients (20.2%) had persistent loco-regional disease (L-R group). The remaining six patients (7.6%) had advanced disease that was not resectable (inoperable group). However, there were no significant clinicopathologic differences between groups, except in treatment course. External beam radiation therapy (EBRT) was performed in 36 patients, and it was significantly more common in the initial-DM and inoperable groups than in the late-DM and L-R groups (66.7%/66.7% vs. 29.2%/18.8%, *p* = 0.001). The cumulative activity of radioactive (RAI) was highest in the initial-DM group, followed by the late-DM and L-R groups; this difference was statistically significant (450 vs. 310 vs. 100 mCi, *p* = 0.001).

### 2.3. Causes of Death of 79 Patients Who Died of WDTC

Among 79 patients who died of WDTC, 62 patients (78.5%) had PTC and 17 patients (21.5%) had FTC (Table 3). Of the 62 patients with PTC, respiratory failure (32.3%) and airway obstruction (30.6%) were the most common causes of death. Respiratory failure was caused by lung metastasis or malignant pleural effusion. Airway obstruction was caused by asphyxia owing to tumor invasion or massive tumor bleeding with tracheo-esophageal fistula. Of the 17 patients with FTC, various complications due to immobilization arising from bone metastasis (35.3%) were the most common cause of death. Long-term immobilization caused by pathologic fracture or cord compression owing to spine metastasis were associated with aspiration pneumonia or sepsis due to infected sores. There were two cases (11.8%) of respiratory failure, but none of these patients died of airway obstruction. Among 79 patients with WDTC, brain metastasis caused increased intracranial pressure or hemorrhage, leading to death in 7 patients (8.9%).

### 2.4. Overall Survival and Clinical Courses of 79 Patients Who Died of WDTC

Overall survival (OS) was defined the time from diagnosis to death. OS was significantly shorter in the inoperable group [8.5 (3.0–26.0) months], followed by the initial-DM, L-R, and late-DM groups [49.0 (31.0–64.0) vs. 68.5 (24.0–94.0) vs. 93.0 (46.5–116.0) months, *p* < 0.001] (Figure 2A and Appendix A). In the initial-DM group, survival rates at 1 year, 3 years, and 5 years were 90.1%, 66.7%, and 33.3%, respectively, and they decreased more steeply than in the late-DM and L-R groups. In the L-R group, survival rates at 1 year, 3 years, and 5 years were 87.5%, 68.8%, and 56.3%, respectively. In the late-DM group, survival rates at 1 year, 3 years, and 5 years were 100%, 87.5%, and 70.8%, respectively. Nine of 16 patients had airway invasion, and the OS of those patients was 24 (14–60) months, while the OS of the 5 patients without airway invasion was 98 (88.5–120.5) months. All patients with loco-regional disease who died within 24 months had tracheal invasion. The OS of patients with PTC was similar to that of all patients with WDTC (Figure 2B); however, in patients with FTC, most deaths occurred in the initial DM group (*n* = 12, 70.6%) (Figure 2C). OS was comparable when only initial DM patients with PTC and FTC were compared, despite their different clinical courses (Figure 3).

Twenty-four patients were diagnosed with late-DM; 6 patients achieved complete remission and 18 had persistent loco-regional disease, but distant metastases occurred in all of these patients during follow-up. Distant metastases were found at a median of 50 months (31.0–71.5) after initial diagnosis, and patients died at a median of 23 months (7.0–51.0) after detection of distant metastases. The duration from detection of distant metastases to death was significantly shorter in the late-DM group compared with the initial-DM group (23.0 vs. 49.0 months, *p* < 0.01).

### 2.5. Comparison of PTC with FTC According to Clinical Courses

Among 62 patients with PTC, the late-DM group was the largest, comprising 37.1% of patients; this was followed in descending order by the initial-DM (33.9%), L-R (22.6%), and inoperable groups (6.5%). Among 17 patients with FTC, most patients were in the initial-DM group (70.6%), followed by the inoperable (11.8%) and L-R groups (11.8%), and then the late-DM group (5.9%) (Table 4). Among 45 patients with PTC and distant metastases, lung metastasis (55.6%) was the most common, followed in decreasing order by two or multi-organ involvement (22.2%), brain (13.3%), and bone (8.9%). Among 15 patients with FTC and distant metastases, bone metastasis (46.7%) was the most common, followed in decreasing order by two or multi-organ involvement (40.0%) and brain (13.3%) involvement (Table 5).

### 2.6. Comparison of OS According to Distant Metastatic Site

Sixty patients had distant metastases prior to death. Lung metastasis was found in 25 patients (41.7%), bone metastasis in 11 patients (18.3%), lung and bone metastases in 10 patients (16.7%), brain metastasis in 8 patients (13.3%), bone and brain metastases in 1 patient (1.7%), and 3 or more organ metastases in the remaining 5 patients (8.3%).

Among 60 patients with distant metastases, OS was shortest with brain metastasis [41.5 (21.0–63.0)], followed by bone, lung, and multiple metastases [49.0 (34.5–80.0) vs. 62.0 (33.0–105.0) vs. 63.5 (43.0–80.5), *p* = 0.412] (Figure 4).

### 2.7. Comparison of OS According to Time Period

We arbitrarily classified patients into three categories according to initial diagnosis: 1996–2000 (*n* = 27), 2003–2009 (*n* = 36), and 2010–2018 (*n* = 16). Clinicopathological characteristics were not different between the three groups (Appendix A), and there was no significantly difference of overall survival between groups (*p* = 0.128, Appendix A).

## 3. Discussion

Of the 483 patients who died with WDTC, only 79 patients (16.4%) died of WDTC, while the remaining 404 patients (83.6%) died of other causes. A significant number of deaths resulted from various diseases other than WDTC, which is consistent with a previous report [15]. However, the direct causes of death in patients dying of WDTC have seldom been reported. Because WDTC has a favorable and indolent clinical course, it is difficult to track the entire clinical course from diagnosis to death in patients who die of WDTC. Furthermore, several prior studies focused on the clinicopathologic characteristics and immediate cause of death in patients who died of thyroid cancer [16,17,18,19,20,21,22]. However, all of those studies had limitations, such as inclusion of anaplastic thyroid carcinoma, which has a worse outcome [16,18,20], inclusion of WDTC with anaplastic transformation [17], inclusion of a small number of subjects [19,21,23], and failure to analyze PTC and FTC separately [17,19,21,22] despite their different clinical features [14,24].

We focused on the cause of death and overall clinical course in 79 patients who died of WDTC, and separated them into PTC and FTC groups. Given the similarity of their derived origin, PTC and FTC are both classified as WDTC; however, their clinical behaviors are quite different [14]. A previous study reported that the most common initial distant metastasis site was lung in PTC and bone in FTC [25,26]. In this study, we also observed that lung in PTC and bone in FTC were the most common metastasis sites, consist with previous reports. This study revealed that the dominant cause of death was also different between PTC and FTC. Respiratory failure and airway obstruction were the dominant causes of death in PTC, whereas complications due to immobilization arising from bone metastasis were the dominant cause of death in FTC. Cause of death was generally consistent with the site of distant metastasis.

Among 60 patients with distant metastases, brain metastasis was found in 8 patients (13.3%); these patients had the worst prognosis, with a median survival of 41.5 months (compared with 62.0 months in lung, 49.0 in bone, and 63.5 months in multi-organ metastases). In particular, brain metastasis at the time of diagnosis showed the shortest median survival of 25.0 months compared with metastasis in other organs (46.5 months in lung, 49.0 months in bone, and 56.0 months in multi-organ metastases) among patients with initial-DM. Early intervention for brain metastasis, such as radiation therapy or surgery, could be important in increasing survival [27,28,29].

In addition, there was a clear difference in the timing of distant metastases according to histology type. Patients who died of PTC had an equal distribution (initial-DM 33.9%, late-DM 37.1%, L-R 22.6%), while 70.6% of patients who died of FTC had initial-DM. Previous studies showed that death from FTC was two times higher than death from PTC [30], and that distant metastases were more common in patients with FTC [24]. However, Mazzaferri et al. presented a similar prognosis for the two histologic types when there was no distant metastasis [5]. In this study, PTC and FTC patients in the initial-DM group had similar OS. Thus, the higher mortality rate in FTC could be owing to a higher rate of distant metastasis.

OS in initial-DM was significantly shorter than that in late-DM (49.0 vs. 93.0, *p* < 0.001); indeed, the late-DM group had a significantly shorter period from detection of metastasis to death compared with the initial-DM group (23.0 vs. 49.0 months, *p* < 0.01). A previous study reported that symptomatic distant metastasis patients had significantly lower survival than asymptomatic patients [31], which might be related to lead-time bias. Sixteen of 33 initial-DM patients (48.5%) and 15 of 24 late-DM patients (62.5%) had symptoms related to distant metastases when these distant metastases were found. The mean (standard deviation) duration from the detection of distant metastasis to death in the initial-DM group with and without symptoms was 44 (25) months and 57 (32) months, respectively. However, the mean (standard deviation) duration from the diagnosis of distant metastasis to death in the late-DM group with and without symptoms was 23 (29) months and 49 (29) months, respectively. Furthermore, 6 of 15 patients with symptomatic late-DM (40%) died within four months of distant metastasis. Considering these results, the difference in duration from the time of diagnosis to death between the initial-DM and late-DM groups could be attributed to lead-time bias.

Because of time-dependent changes in diagnosis and treatment, long-term observation studies are bound to be susceptible to ‘time bias’. Thus, we arbitrarily classified patients into three categories according to the time of initial diagnosis, and conducted a survival analysis according to time period. However, there was no difference of overall survival according to time period. The incidence of thyroid cancer increased, and previous studies suggested that the cause of this increase is related to use of high-resolution neck ultrasonography and the early detection of small PTCs [4,32]. In fact, the increase in prevalence was mostly owing to the increase in proportion of papillary thyroid microcarcinoma, and recurrence and mortality rates were improved [33,34]. Recently, tyrosine kinase inhibitor (TKI) was used in advanced WDTC patients; however, the basic concept of treatment strategy, which consists of surgery for primary lesion followed by radioactive iodine therapy, has not changed. It seems that overall survival of patients who died of WDTC did not change over time.

The interval from the last RAI treatment to death was 21.5 (15.0–40.0), 33.0 (13.0–57.0), and 70.5 (26.5–101.0) months for initial-DM, late-DM, and L-R, respectively. In the L-R group, median cumulative RAI activity was 100 mCi; 4 of 16 patients did not receive RAI owing to advanced disease and poor general condition, and 5 of 16 patients received only 100–200 mCi for the same reasons.

This study had some limitations. First, the cause of death often was inaccurate or unconvincing. To overcome this problem, we reviewed the medical records of all patients who died with WDTC in detail. Second, the pathologic diagnoses of 592 patients were WDTC at the time of the initial diagnosis; however, the initial diagnosis may bias the result because the pathologic diagnosis has evolved during the study period. To overcome this problem, we reviewed the initial histological slide of 79 enrolled patients by experienced pathologic specialist, and confirmed that the selected WDTC cohort consisted of only PTC and WDTC. Third, OS in each group could be affected by lead-time bias, because some patients had symptoms at the initial diagnosis, while others did not. However, this bias is inevitable because of the retrospective nature of this study. Although we tried to correct this problem following established methods [35], this bias was difficult to correct for the following reasons: (1) there is an unknown constant time parameter representing the interval between screen-detectable disease and symptomatic disease for lead-time bias in WDTC; (2) even if the estimated constant value is assumed, it may be inaccurate because of a small sample size, and different constant parameters should be estimated for each group; and (3) it is difficult to satisfy the methodologic assumption that sojourn time must follow an exponential distribution. For these reasons, it is possible that another bias was produced when adjusting lead-time bias. However, to the author’s knowledge, there are no data regarding the dependence of OS on the initial clinical situation. Thus, these results provide information about OS for patients depending on the initial clinical situation.

## 4. Materials and Methods

### 4.1. Patients

We identified 592 deceased patients who were treated for WDTC between 1996 and 2018 at Samsung Medical Center. Information on whether the patient died was obtained from the Korean Statistical Information Service (KOSIS). We retrospectively reviewed the medical records of 592 patients and excluded 109 patients with insufficient medical records or an unknown cause of death owing to transfer to another hospital. After all exclusions, we included the remaining 79 patients in this study. Sixty-two patients had PTC, while 17 patients had FTC. Among the PTC patients, 57 had classic PTC and 5 had follicular-variant PTC. Patients with coexisting poorly differentiated or anaplastic thyroid carcinoma or anaplastic transformation were not included in this study. This study was approved by the Institutional Review Board of Samsung Medical Center (SMC 2019-07-004).

Depending on the clinical course and extent of disease, all 79 patients were categorized into four groups. The first group included six patients who did not undergo surgery because of severe cancer progression (inoperable group). The second group included 33 patients with distant metastases detected within 6 months of diagnosis or just after initial 131-I therapy (initial-DM group). The third group included 24 patients with distant metastases detected during follow-up (late-DM group). The fourth group included 16 patients with persistent loco-regional disease (L-R group).

### 4.2. Treatment and Follow-Up

The primary lesions of all patients were pathologically confirmed. Sixty patients presented with distant metastases; 53% of these distant lesions were detected by computed tomography (CT), while 25% were detected by magnetic resonance image (MRI), 20% by 18F-fluorodeoxyglucose positron emission tomography (PET), 18.3% by pathology (operation or biopsy specimens), 8.3% by diagnostic I-131 scan, and 5% by 99 mTc whole-body bone scintigraphy. Detection of distant metastasis was confirmed by more than one diagnostic tool.

All but six patients underwent total thyroidectomy with curative intent. Prophylactic central lymph node dissection was routinely performed when patients underwent total thyroidectomy. Lateral lymph node dissection was performed with a compartment-based approach in the presence of radiologically or clinically apparent lymph node disease. An initial activity of 30–200 mCi of 131-I was given for remnant ablation or treatment of remaining or metastatic disease, and additional 131-I was given 6–12 months later in patients with any 131-I uptake indicating residual disease. All patients were treated with levothyroxine to suppress thyroid stimulating hormone (TSH) to <0.10 mU/L, and then were followed up with clinical assessment and serum TSH, thyroglobulin (Tg), and anti-Tg antibody measurement every 3 to 6 months. The frequency of follow-up visits and imaging tests was based on the clinical course of the disease. Among 60 patients presenting with distant metastases, metastatic lesions were treated with high dose radioactive (RAI) (88.3%), external beam radiation therapy (53.3%), palliative or curative operation for the metastatic lesion (23.3%), gamma-knife surgery (6.7%), vessel embolization (3.3%), and/or chemotherapy (1.7%), and three patients received tyrosine kinase inhibitor (TKI) treatment (3.8%).

### 4.3. Statistical Analysis

Continuous variables are presented as the mean and standard deviation or the median and interquartile range, as appropriate, and the t-test or Mann–Whitney U test were performed for comparison between groups. Categorical variables are presented as the numbers and percentages. The Chi-square test and Fisher’s exact test were used to compare the clinicopathologic characteristics between groups. The Kruskal–Wallis test was performed with post-hoc analysis using the Mann–Whitney test with the Bonferroni correction to analyze cumulative activity of RAI. The Kaplan–Meier method was used to calculate OS, and groups were compared with the log-rank test. Statistical analysis was performed using SPSS version 25.0 for Windows (IBM, Chicago, IL, USA).

## 5. Conclusions

The cause of death in WDTC patients differed according to metastatic sites. OS differed according to clinical course, but not histologic type in patients who died of WDTC. Timing and the sites of distant metastasis differed between PTC and FTC.

## Figures and Tables

**Figure 1 cancers-12-02323-f001:**
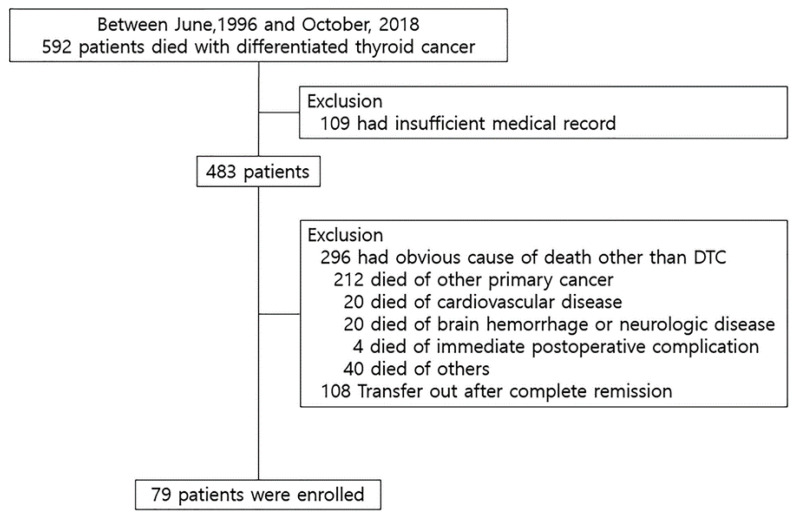
Study populations. DTC, differentiated thyroid carcinoma.

**Figure 2 cancers-12-02323-f002:**
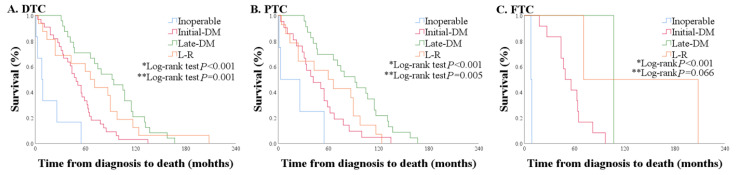
Overall survival according to clinical course in patients with (**A**) all well-differentiated thyroid carcinoma (WDTC), (**B**) papillary thyroid carcinoma (PTC) only, and (**C**) follicular thyroid carcinoma (FTC) only (* comparison of the inoperable, initial distant metastasis (DM), late-DM, and L-R groups, ** comparison of the initial-DM, late-DM group, and L-R groups).

**Figure 3 cancers-12-02323-f003:**
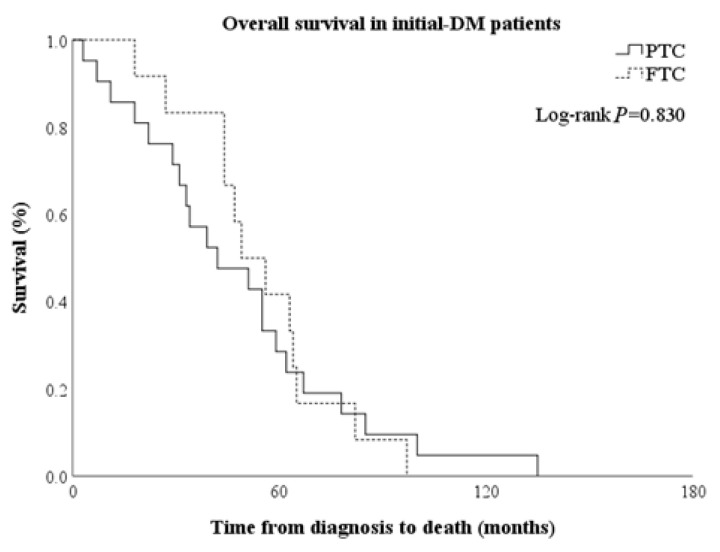
Comparison of overall survival between PTC and FTC in patients with initial-DM.

**Figure 4 cancers-12-02323-f004:**
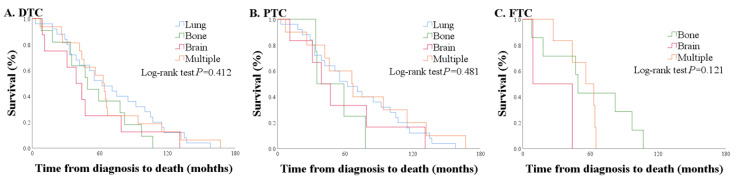
Comparison of overall survival according to distant metastasis site. (**A**) All WDTC, (**B**) PTC only, and (**C**) FTC only.

**Table 1 cancers-12-02323-t001:** Causes of death in 483 patients who died with well-differentiated thyroid carcinoma.

Causes of Death	*n*
Other primary cancer	212
Hepatobiliary cancer	41
Gastrointestinal cancer	38
Lung cancer	35
Breast cancer	26
Hematologic malignancy	18
Gynecologic cancer	13
Urogenital cancer	10
Brain tumor	9
Head and neck cancer	8
Other malignancy ^†^	14
Cardiovascular disease, heart failure	20
Brain hemorrhage, neurologic disease	20
Died of postoperative complications	4
Other disease ^‡^	40
Transfer out after remission (unknown exact cause of death)	108
Total	404

*n*, number of patients; WDTC, well-differentiated thyroid carcinoma; ^†^ other malignancy: thymic cancer, malignant mesothelioma, ewing sarcoma, malignant melanoma, sarcoma, peritoneal carcinoma, metastasis of unknown origin; ^‡^ other disease: chronic kidney disease, asthma, pneumonia, infection, liver failure, rheumatoid disease, chronic pulmonary disease, and accident.

**Table 2 cancers-12-02323-t002:** Clinicopathologic characteristics of 79 enrolled well-differentiated thyroid cancer patients.

Characteristics		Inoperable *n* (%)	Initial-DM *n* (%)	Late-DM *n* (%)	L-R Group *n* (%)	*p*-Value (A vs. B vs. C)
Number of patients		6 (7.6)	33 (41.8)	24 (30.4)	16 (20.3)	
Sex	Female	6 (100)	21 (63.6)	13 (54.2)	12 (75)	0.407
Age	<55	1 (16.7)	9 (27.3)	3 (12.5)	3 (18.8)	0.387
	55 or more	5 (83.3)	24 (72.7)	21 (87.5)	13 (81.3)	
Primary tumor size ^†^	<4 cm	3 (50)	24 (72.7)	21 (87.5)	11 (68.8)	0.298
	4 cm or more	3 (50)	9 (27.3)	3 (12.5)	5 (31.3)	
pT	1	-	5 (15.2)	3 (12.5)	1 (6.3)	0.445
	2	-	4 (12.1)	1 (4.2)	4 (25.0)	
	3	-	15 (45.5)	9 (37.5)	6 (37.5)	
	4	-	9 (27.3)	11 (45.8)	5 (31.3)	
pN	Not assessed	-	1 (3.0)	1 (4.2)	1 (6.3)	0.918
	N0	-	15 (45.5)	9 (37.5)	8 (50)	
	N1a	-	4 (12.1)	5 (20.8)	3 (18.8)	
	N1b	-	13 (39.4)	9 (37.5)	4 (25.0)	
Lymphovascular invasion	Yes	-	14 (42.4)	3 (12.5)	2 (12.5)	0.015
Resection margin	Positive	-	9 (27.3)	9 (37.5)	2 (12.5)	0.221
Extent to central LN	Yes	-	14 (42.4)	11 (45.8)	6 (37.5)	0.872
Extent to lateral LN	Yes	-	13 (39.4)	9 (37.5)	4 (25.0)	0.598
Extent of surgery	Total	-	31 (93.9)	23 (95.8)	14 (87.5)	0.576
	Less than total	-	2 (6.1)	1 (4.2)	2 (12.5)	
EBRT	Yes	4 (66.7)	22 (66.7)	7 (29.2)	3 (18.8)	0.001
Cumulative dose of RAI		0	450 (175–815)	310 (280–707.5)	100 (25–207.5)	0.001 *

A, initial distant metastasis (DM) group; B, late-DM group; C, loco-regional disease (L-R) group. * The Kruskal–Wallis test was performed with post-hoc analysis using the Mann–Whitney test with Bonferroni correction. A vs. C (*p* < 0.001); B vs. C (*p* = 0.001). Categorical variables are shown as number (percentage) and continuous variables are shown as median (interquartile range). *n*, number of patients; Not assessed, lymph nodes were not assessed during surgery; LN, lymph node; Total, total thyroidectomy; EBRT, external beam radiation therapy; RAI, radioactive iodine. †Primary tumor size in inoperable group was measured by thyroid ultrasound.

**Table 3 cancers-12-02323-t003:** Immediate cause of death.

Cause of Death	WDTC (*n* = 79) *n* (%)	PTC (*n* = 62) *n* (%)	FTC (*n* = 17) *n* (%)
Respiratory failure	22 (27.8)	20 (32.3)	2 (11.8)
Airway obstruction	19 (24.1)	19 (30.6)	0
Complications due to immobilization	11 (13.9)	5 (8.1)	6 (35.3)
Brain metastasis	7 (8.9)	4 (6.5)	3 (17.6)
Cachexia	6 (7.6)	3 (4.8)	3 (17.6)
Liver failure due to hepatic metastasis	2 (2.5)	1 (1.6)	1 (5.9)
Unclear	12 (15.2)	10 (16.1)	2 (11.8)

*n*, number of patients; WDTC, well-differentiated thyroid carcinoma; PTC, papillary thyroid carcinoma; FTC, follicular thyroid carcinoma.

**Table 4 cancers-12-02323-t004:** Comparison of PTC with FTC according to clinical course.

Characteristics		PTC (*n* = 62), *n* (%)					FTC (*n* = 17), *n* (%)				
		Inoperable	Initial-DM Group	Late-DM Group	L-R Group	*p*-Value (A vs. B vs. C)	Inoperable	Initial-DM Group	Late-DM Group ^‡^	L-R Group ^‡^	*p*-Value (A vs. B vs. C)
Number of patients		4 (6.5)	21 (33.9)	23 (37.1)	14 (22.6)		2 (11.8)	12 (70.6)	1 (5.9)	2 (11.8)	
Sex	Female	4 (100.0)	9 (42.9)	13 (56.5)	11 (78.6)	0.112	2 (100.0)	12 (100.0)	0	1	0.02
Age	<55	1 (25.00)	6 (28.6)	3 (13.0)	2 (14.3)	0.439	0 (0.0)	3 (25.0)	0	1	>0.99
	55 or more	3 (75.0)	15 (71.4)	20 (87.0)	12 (85.7)		2 (100.0)	9 (75.0)	1	1	
Primary tumor size ^†^	<4 cm	1 (25.00)	13 (61.9)	20 (87.0)	10 (71.4)	0.160	2 (100.0)	11 (91.7)	1	1	0.554
	4 cm or more	3 (75.0)	8 (38.1)	3 (13.0)	4 (28.6)		0 (0.0)	1 (8.3)	0	1	
pT	1	-	2 (9.5)	3 (13.0)	1 (7.1)	0.371	-	3 (25.0)	0	0	>0.99
	2	-	1 (4.8)	0 (0.0)	3 (21.4)		-	3 (25.0)	1	1	
	3	-	11 (52.4)	9 (39.1)	5 (35.7)		-	4 (33.3)	0	1	
	4	-	7 (33.3)	11 (47.8)	5 (35.7)		-	2 (16.7)	0	0	
pN	Not assessed	-	0 (0.0)	1 (4.3)	1 (7.1)	0.430	-	1 (8.3)	0	0	>0.99
	N0	-	4 (19.0)	8 (34.8)	6 (42.9)		-	11 (91.7)	1	2	
	N1a	-	4 (19.0)	5 (21.7)	3 (21.4)		-	0	0	0	
	N1b	-	13 (61.9)	9 (39.1)	4 (28.6)		-	0	0	0	
Lymphovascular invasion	Yes	-	6 (28.6)	2 (8.7)	1 (7.1)	0.177	-	8 (66.7)	1	1	>0.99
Resection margin	Positive	-	8 (38.1)	9 (39.1)	2 (14.3)	0.239	-	1 (8.3)	0	0	>0.99
Extent to central LN	Yes	-	14 (66.7)	11 (47.8)	6 (42.9)	0.301	-	0	0	0	-
Extent to lateral LN	Yes	-	13 (61.9)	9 (39.1)	4 (28.6)	0.118	-	0	0	0	-
Extent of surgery	Total	-	21 (100.0)	22 (95.7)	13 (92.9)	0.708	-	10 (83.3)	1	1	0.516
	Less than total	-	0 (0.0)	1 (4.3)	1 (7.1)		-	2 (16.7)	0	1	

A, initial-DM group; B, late-DM group; C, L-R group, Categorical variables are shown as number (percentage) and continuous variables are shown as median (interquartile range). *n*, number of patients; Not assessed, lymph nodes were not assessed during surgery; LN, lymph node; Total, total thyroidectomy. A, initial-DM group; B, late-DM group; C, L-R group; PTC, papillary thyroid carcinoma; FTC, follicular thyroid carcinoma. ^†^ Primary tumor size in inoperable group was measured by thyroid ultrasound. ^‡^ Because of the small number of enrolled patients, we did not calculate percentages in the late-DM and L-R groups in FTC patients.

**Table 5 cancers-12-02323-t005:** Timing and site of distant metastasis in PTC and FTC patients.

Site of Distant Metastasis	WDTC (*n* = 60) (*n*)				PTC (*n* = 45) (*n*)				FTC (*n* = 15) (*n*)			
	Total (*n*, %)	Inoperable	Initial-DM Group	Late-DM Group	Total Cases in PTC (*n*, %)	Inoperable	Initial-DM Group	Late-DM Group	Total Cases in FTC (*n*, %)	Inoperable	Initial-DM Group	Late-DM Group
Lung	25 (41.7)	0	14	11	25 (55.6)	0	14	11	0	0	0	0
Bone	11 (18.3)	1	8	2	4 (8.9)	0	3	1	7 (46.7)	1	5	1
Lung + Bone	10 (16.7)	0	7	3	5 (11.1)	0	2	3	5 (33.3)	0	5	0
Brain	8 (13.3)	1	3	4	6 (13.3)	0	2	4	2 (13.3)	1	1	0
Bone + Brain	1 (1.7)	1	0	0	1 (2.2)	1	0	0	0	0	0	0
3 & more organs	5 (8.3)	0	1	4	4 (8.9)	0	0	4	1 (6.7)	0	1	0
Total cases	60	3	33	24	45	1	21	23	15	2	12	1

*n*, number of patients; WDTC, well-differentiated thyroid carcinoma; PTC, papillary thyroid carcinoma; FTC, follicular thyroid carcinoma.

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
