# Peer review of "Clinical Course from Diagnosis to Death in Patients with Well-Differentiated Thyroid Cancer"

_cancers, 2020, doi:10.3390/cancers12082323_

Round 1

Reviewer 1 Report

This is an interesting study, which addressed a clinical course from diagnosis to death (mainly death of disease) in a cohort of patients with differentiated thyroid cancer (DTC). Even dataset was collected from a single institution, a number of analyzed cases is similar to the annual death of thyroid cancer in Korea (as per Globocan estimates),

MAJOR

  • The study cohort was enrolled over 22y period (1996-2018), which encompassed several editions of the international and local management guidelines = changing diagnostic modalities, treatment options, and follow-up strategies. No doubt that patients could be managed differently in mid-90s vs. late 2010s, which influence outcome trends. This issue should be carefully addressed in a separate section of Results (e.g. stratification by decades or shorter interval, such 7y or 5y periods).
  • Lack of pathological validation of the initial diagnosis may bias results because of continuously evolving diagnostic criteria. Ideally, histological slides of the primary tumors should be retrieved and reviewed; otherwise this issue should be clearly indicated as the major limitation of the study.
    • For example, there were 2/20 cases of FTC with nodal metastasis (Table 2), which is counterintuitive and these cases most likely were PTCs of follicular variant.
    • Furthermore, the authors emphasized that “inclusion of DTC with anaplastic transformation” was a significant limitation of previous publications; however, this concept (minor anaplastic component/foci in DTC) simply didn’t exist in 1990s and early 2000s. In the same sense, diagnostic criteria of poorly differentiated carcinoma has changed significantly. Therefore, there is no confidence in the purity of the selected DTC cohort without adequate slides review by expert pathologist(s).
  • Progress of advanced DTC is biologically linked to development of RAI resistance; however, the authors completely ignored this issue. How many patients in each subgroup developed RAI-refractory disease? Mean time? Influence on outcome?

MINOR

  • It would be of immense help for readers if the authors could provide chart/graph of timeline in each group, showing the main points (diagnosis, recurrence, metastasis, death, etc.) and duration.
  • How many patients didn’t receive adequate treatment, e.g. completely refused treatment or refused radical treatment?
  • Some experts consider poorly differentiated thyroid carcinoma (PDTC) as a subtype of DTC; to avoid a potential confusion, the authors may opt choosing a less ambiguous term, WDTC (well-diff thyroid carcinoma) instead of DTC in their manuscript.
  • Table 1: sort other primaries by descending frequency.
  • Stylistic: rounded-in-rounded brackets (lines 153-155) and rounded-in-squared brackets (lines 107-108) should be replaced with squared-in-rounded brackets, i.e. [txt (txt) txt]

Reviewer 2 Report

In the present work the authors analyze a retrospective cohort of patients diagnosed with differentiated thyroid cancer (DTC), including both papillary thyroid carcinoma (PTC) and follicular thyroid carcinoma (FTC). The authors conclude that only a small percentage of DTC patients died from the disease (16%), while the other patients died from other causes. Furthermore, disease presenting with metastasis correlates with poorer prognosis.  

I have a few minor comments:

  • The authors mentioned (lane 57) that 212 patients died of other advanced primary cancers. They should explain a little more about how is this determined. Would then be DTC a potential metastatic site from other cancer types?
  • Labels on graphs in figures 2 and 4 are hard to see.
  • The x-axis on figure 3 must be labelled as “months” instead of “month”
  • Table 5 must be included within one page

Round 2

Reviewer 1 Report

p. 11, line 257: "with only PTC and WDTC" should be "with only PTC and FTC".